# Awareness of anaphylaxis among public in Al-Ahsa City of Saudi Arabia: A cross-sectional study

Ahmed Alanazy [1,2,3] *, Raja Saad Boodai[1,2,3], Badiah Ibrahim Alhulaybi[1,2,3], Amenah Ibrahim Alraihan[1,2,3], Raghad Ahmed Almulhim[1,2,3], Bainah Fahad Almulhim[1,2,3], Suchithra K. Rajappan[2,3,4], Abdullah Alruwaili[1,2,3], Ahmad Alanazi[5]

1 Emergency Medical Services Program, College of Applied Medical Sciences, King Saud Bin Abdulaziz University for Health Sciences, Al Ahsa, Saudi Arabia, 2 King Abdullah International Medical Research Center, Al Ahsa, Saudi Arabia, 3 Ministry of National Guard - Health Affairs, Al Ahsa, Saudi Arabia, 4 Basic Science Department, College of Science and Health Professions, King Saud Bin Abdulaziz University for Health Sciences, Al Ahsa, Saudi Arabia, 5 Respiratory Service, King Abdulaziz Medical City, Riyadh, Saudi Arabia

* alanazyah@gmail.com

**Data Availability Statement:** The data have been uploaded to the Harvard Dataverse. You can access it via the following link: Alanazy, Ahmed, 2024, "Awareness of Anaphylaxis Among Public in

## Abstract

### Background

Anaphylaxis is a severe, life-threatening allergic reaction requiring prompt treatment with epinephrine. However, gaps in public understanding exist globally. To guide future education efforts, this study assessed anaphylaxis awareness among adults in Al-Ahsa, Saudi Arabia.

### Methods

A cross-sectional survey assessed knowledge among 380 adults using a validated questionnaire. Convenience sampling was used with data collected via online platforms. Descriptive statistics were calculated, and associations were tested. We analyzed the data using SPSS software.

### Results

Knowledge was suboptimal, with a mean score of 34.5/60. Most identified the anaphylaxis definition (76.1%) and rapid symptom onset (78.9% within minutes). But just 37.9% recognized epinephrine as a first-line treatment and only 17.4% had auto-injector training. Nearly half (48.4%) needed to learn the proper EpiPen injection site. Under half understood the need for long-term allergen avoidance. Older age, more education, and Job status were associated with higher scores.

### Conclusion

This study reveals critical gaps in anaphylaxis knowledge among adults in Al-Ahsa, Saudi Arabia. Poor understanding of emergency treatment and long-term management highlights the urgent need for improved public awareness. Targeted educational initiatives must

Al-Ahsa City of Saudi Arabia: A Cross-sectional Study", https://doi.org/10.7910/DVN/SUAK5A, Harvard Dataverse, V1.

**Funding:** The author(s) received no specific funding for this work.

**Competing interests:** The authors have declared that no competing interests exist.

emphasize prompt epinephrine administration, proper auto-injector use, and allergen avoidance. Collaborative efforts engaging healthcare professionals, schools, and policymakers are essential to enhance anaphylaxis preparedness through focused training and tailored awareness campaigns. This will empower the Saudi public to recognize symptoms quickly and respond effectively, saving lives from this dangerous allergic reaction.

## Introduction

Life-threatening allergic reactions, known as anaphylaxis, can happen seconds after contact with a trigger such as food, medicine, or an insect sting [1, 2]. Anaphylaxis needs immediate treatment with an epinephrine injection to reverse dangerous symptoms and prevent worsening [3, 4]. While many things can cause anaphylaxis, some of the most common triggers are peanuts, tree nuts, shellfish, eggs, and milk [5, 6]. Studies show that anaphylaxis is increasing around the world [7, 8]. Estimates report 50 to 112 episodes per 100,000 people [9, 10].

In Saudi Arabia, the prevalence of anaphylaxis among emergency department admissions was 0.00026% in one study [11]. Most cases were children and young adults. Anaphylaxis can sometimes lead to long-term anxiety or post-traumatic stress disorder [12]. This might create excessive caution around objectively safe foods or situations. Knowing and avoiding specific allergies is the best prevention.

Even though anaphylaxis is serious, research in Western countries has found gaps in public understanding [13]. Little information exists on anaphylaxis awareness in the Middle East, although food allergies are rising in the region [14, 15]. Evaluating what the general public knows about anaphylaxis is an important first step. It can guide education efforts to improve recognition and response [1].

Al-Ahsa has over 1 million residents in both urban and rural areas [16]. This diversity makes it useful for measuring anaphylaxis understanding across different groups. We expect many residents will be unable to identify or manage anaphylaxis correctly.

We aim to assess anaphylaxis knowledge among adults in Al-Ahsa, Saudi Arabia. Identifying specific gaps can highlight where increased education is needed to improve anaphylaxis preparedness in this region. This study will provide insights into the current level of anaphylaxis understanding in Al-Ahsa.

## Method

### Study design and setting

This cross-sectional study assessed awareness of anaphylaxis among the public in Al-Ahsa City in Saudi Arabia. Data was collected via an online survey distributed through social media platforms, including Google Forms, Twitter, WhatsApp, Snapchat, and Facebook. This cross-sectional study was conducted from 17 October 2022 to 17 October 2023.

### Study subjects

The study population included adult men and women aged 18–80 years residing in the Al-Ahsa region of Saudi Arabia. Convenience sampling was used to recruit participants. The inclusion criteria were: 1) adults aged 18–80 years, 2) residing in the Al-Ahsa region, and 3) willing to participate in the study. Those aged below 18 or over 80, living outside Al-Ahsa, or unwilling to participate were excluded.

### Sample size

The required sample size was calculated using the single population proportion formula. Assuming 50% awareness with a 5% margin of error and a 95% confidence interval, the minimum required sample size was 380. This was increased by 10% to account for non-responses, giving a target sample size of 418.

### Data collection

Data was collected using a pre-structured, close-ended questionnaire administered through the online survey. The questionnaire collected information on demographic variables such as age, gender, education level, nationality, and social status. It also contained questions assessing knowledge and awareness related to anaphylaxis.

The questionnaire was validated by expert review and pilot testing on 10% of the sample before full administration. Reliability was assessed using Cronbach's alpha. Any issues identified during piloting were addressed before the full deployment of the survey.

### Data analysis

Completed surveys were checked for completeness and accuracy. Data analysis was conducted using statistical SPSS software. Descriptive statistics such as frequencies, percentages, means, standard deviations, and medians were calculated as appropriate based on the type and distribution of variables. Associations were tested using appropriate statistical tests. P-values $<0.05$ were considered statistically significant.

### Ethical considerations

Ethical approval was obtained from the king Abdullah international medical research center -view board before the study commenced. The approval code is NRA22A/026/08. Every participant's informed consent was diligently acquired before their involvement in the study. Throughout the research, we preserved the confidentiality and anonymity of all participants' data.

## Results

After data were extracted from the online survey, it was revised, coded, and fed to statistical software IBM SPSS version 22(SPSS, Inc. Chicago, IL). Based on inclusion and exclusion criteria for the study data of 380 respondents was included for statistical analysis. Majority of respondents were in the age group of 18-25years (65.8%) in which 271 (71.3%) of them were females and 109 (28.7%) were male. A higher number of respondents were holding bachelor's degree (236, 69.2%). Almost 372 (97.9%) of the participants were Saudis and 65.8% of them were single. There was a statistically significant difference in the anaphylaxis awareness score by age, educational level and the job status of study participants. The mean score of bachelor's degree was significantly lower than those of the participants with lower educational levels. Also, there is a significant difference in mean awareness score for the participants aged above 26 years and above 50 years of age. The detailed information about the demographic characteristics and the mean score of anaphylaxis awareness differences among the study subjects is presented in Table 1.

Table 2 shows that most of the participants answered that anaphylaxis is a potentially life-threatening reaction to a trigger (76.1%). 78.9% answered that the symptoms of anaphylaxis that can occur minutes after contact however, only 2.9% of the study participants reported that symptoms can occur in a day after contact. 68.9% of them chose that exercise isn't likely to

**Table 1. Demographic characteristics and anaphylaxis awareness differences among respondents.**

| Demographic characters | | N (%) | Awareness Score | t/F |
| --- | --- | --- | --- | --- |
| | | | Mean ± SD | (p) |
| Age | 18–25 | 250 (65.8) | 14.67±3.25 | 2.787 (0.041)* |
| | 26–36 | 72 (18.9) | 15.57±3.55 | |
| | 37–49 | 31 (8.2) | 14.26±3.16 | |
| | 50 or above | 27 (7.1) | 15.96 ±2.65 | |
| Gender | Female | 271 (71.3) | 14.99±3.36 | 0.856 (0.392)NS |
| | Male | 109 (28.7) | 14.67±3.10 | |
| Nationality | Saudi | 372 (97.9) | 14.88 ±3.30 | .400 (0.527)NS |
| | Non-Saudi | 8 (2.1) | 15.62±2.45 | |
| Education level | Intermediate | 4 (1.1) | 18.25±3.30 | 2.820 (.025)* |
| | High school | 71 (18.7) | 15.04 ±3.24 | |
| | Diploma | 20 (5.3) | 16.35±2.93 | |
| | Bachelors | 263 (69.2) | 14.79 ±3.29 | |
| | Graduate studies | 22 (5.8) | 13.77±3.16 | |
| Marital status | Single | 250 (65.8) | 14.76±3.20 | 1.371 (.251)NS |
| | Married | 123 (32.4) | 15.06 ±3.46 | |
| | Divorced | 6 (1.6) | 17.00±2.61 | |
| | Widow | 1(0.3) | 18.00±. | |
| Job Status | Yes | 173(45.5) | 13.70±2.72 | 47.391 (0.000)* |
| | No | 207(54.5) | 15.90±3.38 | |

$P < .05$ * Significant at 5% Level, NS: Non-significant.

cause anaphylaxis and 5.3% of the respondents answered that insects stinging is not likely to cause anaphylaxis. A total of 37.9% the first line treatment of anaphylaxis is epinephrine whereas only 4.5% represents salbutamol inhaler. From this we can conclude that there is adequate knowledge about the term anaphylaxis since the participants are educated and knows about the terminology as there is a chances for hearing about anaphylaxis from their reading, television, social media etc.

Table 3 represents that majority of the participants (51.3%) doesn't hear about self-injection EpiPen. EpiPen is an auto-injector that contains epinephrine, a medication that can help decrease your body's allergic reaction. The question is that you must wait for symptoms of anaphylaxis to appear before giving self-injection EpiPen some people which represents 34.7% chose the answer neither while only 5.0% chose the answer strongly disagree. Most of them 43.98% chose the answer neither for the question that if you should remove a person's clothes before giving a self-injection EpiPen most of them 43.98% chose the answer neither, however, just 4.97% chose the answer strongly disagree. 48.4% of the respondents are unaware about the site where should EpiPen inject however, a few people answered buttock 7.1%. From this it's clearly understood that the participants lack awareness about in using the EpiPen for anaphylaxis. There is a statistically significant difference in the mean awareness score ($p<0.05$) about using EpiPen for anaphylaxis.

Table 4 Majority of the participants believe that medications and vaccines can cause an anaphylaxis reaction in the body and 62.9 of the respondents chose false as their opinion about if they are not at risk for a life-threatening reaction in the future if they had a mild allergic reaction to an allergen in the past.

Table 5 There is a statistically significant relation is there for the public knowledge about the management of the anaphylaxis condition and anaphylaxis reaction ($p<0.05$). Participants

**Table 2. Participants opinion for the questions regarding the for knowledge about anaphylaxis.**

| Questions regarding the for knowledge about anaphylaxis | | No | % |
|---|---|---|---|
| Definition of anaphylaxis | A condition in which your airways narrow and swell may produce extra mucus | 46 | 12.1 |
| | Severe and potentially life-threatening reaction to a trigger | 289 | 76.1 |
| | A condition that caused by excess fluid in the lungs | 7 | 1.8 |
| | I don't know | 38 | 10.0 |
| Most obvious symptom of anaphylaxis is a rash | Strongly agree | 132 | 34.7 |
| | Agree | 151 | 39.7 |
| | Neither | 64 | 16.8 |
| | Disagree | 25 | 6.6 |
| | Strongly disagree | 8 | 2.1 |
| Symptoms of anaphylaxis can occur | Minutes | 300 | 78.9 |
| | Hours | 65 | 17.1 |
| | Days | 11 | 2.9 |
| | Either | 4 | 1.1 |
| Which of the following is not likely to cause anaphylaxis | Medications | 38 | 10.0 |
| | Vaccines | 26 | 6.8 |
| | Latex | 34 | 8.9 |
| | Exercise | 262 | 68.9 |
| | Insects | 20 | 5.3 |
| What is the first line in treatment of anaphylaxis | Give epinephrine injection | 144 | 37.9 |
| | Call ambulance | 72 | 18.9 |
| | Salbutamol inhaler | 17 | 4.5 |
| | Antihistamine injection | 34 | 8.9 |
| | I don't know | 113 | 29.7 |

are aware about the how to manage the situation if there is an allergic reaction occurs.45.2% of the respondents opinioned that they need to avoid substances that trigger anaphylaxis symptoms.

In this survey, 11% of participants diagnosed with anaphylaxis due to various causes were admitted to an observation unit or hospital. In contrast, the majority (89%) of respondents had not been diagnosed with anaphylaxis or received any medical treatment (Fig 1).

Table 6 shows the overall average awareness score of anaphylaxis among the Public in Al-Ahsa City was 34.50. Higher scores indicated higher awareness of anaphylaxis is realted with the particiapnts higher educational levels, job status and the age, in which the total awareness score about anaphylaxis among the public is less than the mean score of 50. More measures like educational campaigns and programs about anaphylaxis advertisement through mass media and social networks, needs to be implemented to create more knowledge among public. In addition, health policies regarding anaphylaxis can be implemented among public, such as providing with epinephrine injections, labeling who have allergies, and training people to prevent serious complications.

## Discussion

This cross-sectional study assessed anaphylaxis awareness among 380 adults in Al-Ahsa, Saudi Arabia. Knowledge was suboptimal, with a mean score of 34.5/60. Most identified the anaphylaxis definition and rapid symptom onset correctly. However, substantial gaps existed in recognizing proper treatment and management. Only 37.9% knew epinephrine is first-line, and

**Table 3. Public knowledge about using EpiPen for anaphylaxis.**

| Questions about using EpiPen for anaphylaxis | | No (%) | Awareness Score | t/F |
|---|---|---|---|---|
| | | | Mean ± SD | (p) |
| Ever heard about self-injection EpiPen? | Yes | 139 (36.6) | 11.28±2.46 | 77.630 (0.00)* |
| | No | 195 (51.3) | 13.91±1.79 | |
| | Maybe | 46 (12.1) | 14.39±1.79 | |
| Ever received instructions on how to use the self-injection EpiPen | Yes | 68(17.4) | 10.50 ±2.416 | 56.137 (0.00)* |
| | No | 277(72.9) | 13.55±2.115 | |
| | Maybe | 35(9.2) | 13.63±1.91 | |
| Wait for symptoms of anaphylaxis to appear before giving the self-injection EpiPen | Strongly agree | 73 (19.2) | 10.36±2.28 | 54.185 (0.00)* |
| | Agree | 109 (28.7) | 12.57±2.01 | |
| | Neither | 132 (34.7) | 14.16±1.81 | |
| | Disagree | 47 (12.4) | 14.28±1.60 | |
| | Strongly disagree | 19 (5.0) | 14.58±2.19 | |
| Site should EpiPen injected | Arm | 57 (15.0) | 10.25±2.01 | 118.998 (0.00)* |
| | Thigh | 112 (29.5) | 11.77±2.25 | |
| | Buttock | 27 (7.1) | 12.78±1.72 | |
| | I don't know | 184 (48.4) | 14.65±1.28 | |
| Remove person's clothes before giving self-injection EpiPen | Strongly agree | 48 (12.6) | 10.04±2.65 | 43.334 (0.00)* |
| | Agree | 88 (23.2) | 12.09±2.08 | |
| | Neither | 167 (43.9) | 14.01±1.88 | |
| | Disagree | 58 (15.3) | 13.97±1.75 | |
| | Strongly disagree | 19 (5.0) | 13.00±2.24 | |
| Self-injection EpiPen should be given early in symptoms of anaphylaxis | True | 331 (87.1) | 12.88±2.46 | 7.566 (0.006)* |
| | False | 49 (12.9) | 13.90±2.16 | |

P < .05 * Significant at 5% Level, NS: Non-significant.

17.4% had auto-injector training. Nearly half did not know the right injection site for EpiPens. Under half understood the need to avoid triggers long-term to prevent recurrences. Older age, more education, and job status were associated with higher scores.

In comparing our findings on anaphylaxis awareness in Al-Ahsa, Saudi Arabia, with previous studies, it becomes evident that gaps in knowledge and preparedness are common in our region. The studies conducted by Patnaik et al. (2020) and Ibrahim et al. (2014) highlighted the importance of knowledge and management skills among educated persons, indicating a moderate understanding but suboptimal management skills [17, 18]. The need for proper

**Table 4. The awareness of allergy complications among community.**

| Questions regarding the awareness of allergy complications | No | % |
|---|---|---|
| Medications and vaccines cause an anaphylaxis reaction. | | |
| Strongly agree | 190 | 50.0 |
| Agree | 123 | 32.4 |
| Neither | 52 | 13.7 |
| Disagree | 15 | 3.9 |
| If you had a mild allergic reaction to an allergen in the past, then you are not at risk for a life-threatening reaction in the future | | |
| TRUE | 141 | 37.1 |
| FALSE | 239 | 62.9 |

**Table 5. The assessment of public knowledge about the management of the allergic and anaphylaxis reaction.**

| The best way to manage anaphylaxis condition | Had a mild allergic reaction | | | | P-value |
| --- | --- | --- | --- | --- | --- |
| | True | | False | | |
| | No | % | No | % | |
| Avoid substances that trigger anaphylaxis symptoms | 47 | 45.2 | 57 | 54.8 | 0.000* |
| Always be sure to carry the self-injection EpiPen | 20 | 66.7 | 10 | 33.3 | |
| Knowing how to use self-injection EpiPen | 4 | 57.1 | 3 | 42.9 | |
| All the above | 70 | 29.3 | 169 | 70.7 | |
| Total | 141 | 37.1 | 239 | 62.9 | |

P < .05 * Significant at 5% Level, NS: Non-significant.

training and continued medical education is underscored, aligning with our findings that older age and higher education among the participants exhibited higher awareness scores. Also, González-Díaz et al. (2021) emphasized the significance of recognizing, diagnosing, and treating anaphylaxis promptly [19]. Interestingly, they found that older age, more education, and job status with more experience demonstrated better knowledge, emphasizing the role of practical experience in improving awareness. This observation resonates with our study, where older age participants exhibited higher awareness scores, likely due to their broader life experiences and allergy exposure. Moreover, the research by Jacobson et al. (2012) shed light on prehospital providers' knowledge gaps in the United States, revealing that while paramedics could recognize classic anaphylaxis, they struggled with atypical presentations and proper treatment protocols [20]. This finding mirrors our study's results, where participants had deficiencies in recognizing anaphylaxis triggers and the correct first-line treatment, particularly using epinephrine auto-injectors.

The study conducted by Cimen et al. (2022) in Turkey highlighted a need for sufficient knowledge among educated participants regarding epinephrine doses and EpiPen prescriptions [21]. Similarly, our study identified gaps in understanding EpiPen usage and the correct site for injection, indicating a global need for comprehensive education on epinephrine

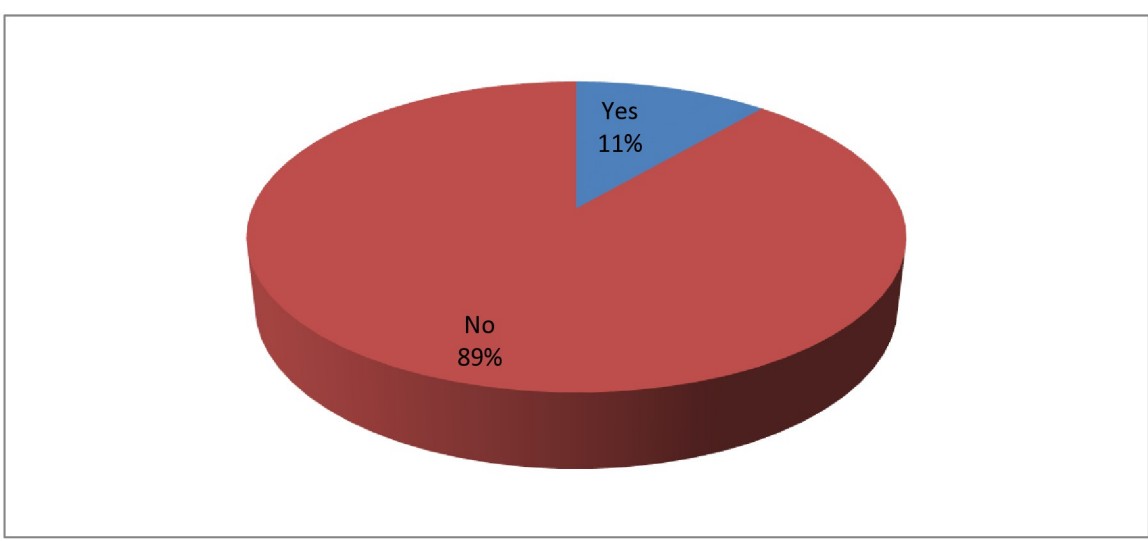

**Fig 1. Propotion of study participants diagnosed with anaphylaxis and their medical treatment outcomes.**

**Table 6. Shows the knowledge score with difference among public's opinions.**

|  | Mean | SD | Std.Error of mean |
|---|---|---|---|
| Awareness score for anaphylaxis | 14.90 | 3.29 | .169 |
| Awareness score for Epi pen usage | 13.01 | 2.45 | .126 |
| Awareness score for management of the anaphylaxis | 6.59 | 0.15 | 0.001 |
| Total awareness in the public | 34.50 | 5.89 | .296 |

administration. Furthermore, research by Alsuhaibani et al. (2019) explored teachers' knowledge and attitudes toward anaphylaxis in Saudi Arabia, revealing a need for more awareness [22]. This finding parallels our study's outcomes, emphasizing the urgency for educational programs among public and the broader community, including teachers who play a crucial role in the safety of adults residents in Alahsa city.

Our findings underline the urgent need for widespread educational initiatives. Implementing targeted awareness programs, including community workshops and school interventions, can empower individuals to recognize anaphylactic symptoms promptly and respond effectively, potentially averting life-threatening situations [9, 23].

Also, our study highlights the necessity for specialized training among the public residence in Al-Ahsa City should undergo continuous education to enhance their knowledge and management skills related to anaphylaxis. This training should focus on recognizing symptoms and properly administering treatments like epinephrine, ensuring a comprehensive approach to anaphylaxis care [19, 24].

Furthermore, our results emphasize the importance of collaboration between the individual healthcare organization and educational institutions awareness programs, role plays, advertisements through mass media and social networks, educational programs, audio-visual training must be implemented to create public awareness. Integrating anaphylaxis education into school curricula ensures that future generations grow up with a solid understanding of this condition, fostering a safer environment for all [25].

Lastly, our study points to the need for policy changes and standardization in anaphylaxis management protocols. By incorporating evidence-based practices into healthcare policies, Saudi Arabia can significantly improve the outcomes for individuals experiencing anaphylaxis [22].

A key strength of this study was the use of a large, diverse sample of 380 adults from the Al-Ahsa region. This provided a comprehensive assessment of anaphylaxis awareness across different demographic groups in the population. Additionally, the questionnaire tool was validated through expert review and pilot testing, supporting the validity of measured knowledge.

However, there were some limitations. The cross-sectional design provided insights at one point in time but could not evaluate changes over time. Convenience sampling introduced potential selection bias, limiting generalizability of findings. The use of online surveys excluded populations without technology access. Self-reported data could not be independently verified.

In conclusion, our study underscores the urgent need for targeted education initiatives. Collaborative efforts involving individual health professionals, educational institutions, and policymakers are essential. Implementing focused awareness programs, integrating anaphylaxis education into schools, and providing specialized training for individuals are vital steps. By addressing these shortcomings, Saudi Arabia can enhance public preparedness, ensuring swift and effective responses to anaphylactic emergencies. This research is a pivotal

foundation, guiding future efforts to create a safer environment for individuals vulnerable to anaphylaxis in Al-Ahsa and beyond.

## Author Contributions

**Data curation:** Raja Saad Boodai, Badiah Ibrahim Alhulaybi, Bainah Fahad Almulhim, Abdullah Alruwaili.

**Formal analysis:** Badiah Ibrahim Alhulaybi, Bainah Fahad Almulhim, Suchithra K. Rajappan, Abdullah Alruwaili, Ahmad Alanazi.

**Investigation:** Amenah Ibrahim Alraihan, Raghad Ahmed Almulhim, Abdullah Alruwaili, Ahmad Alanazi.

**Methodology:** Raja Saad Boodai, Amenah Ibrahim Alraihan, Abdullah Alruwaili.

**Software:** Badiah Ibrahim Alhulaybi.

**Supervision:** Ahmed Alanazy, Abdullah Alruwaili.

**Validation:** Badiah Ibrahim Alhulaybi.

**Writing – original draft:** Ahmed Alanazy, Raja Saad Boodai, Amenah Ibrahim Alraihan, Raghad Ahmed Almulhim.

**Writing – review & editing:** Ahmed Alanazy.

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
