## [Decision Letter · Decision Letter 0]

23 Oct 2024

PONE-D-24-36838Awareness of Anaphylaxis Among Public in Al-Ahsa City of Saudi Arabia: A Cross-sectional StudyPLOS ONE

Dear Dr. Alanazy,

Thank you for submitting your manuscript to PLOS ONE. After careful consideration, we feel that it has merit but does not fully meet PLOS ONE’s publication criteria as it currently stands. Therefore, we invite you to submit a revised version of the manuscript that addresses the points raised during the review process.

We look forward to receiving your revised manuscript.

Kind regards,

Vinh Le Ba, PhD in Pharmaceutical Science

Academic Editor

PLOS ONE

Journal Requirements:

The name of the colleague or the details of the professional service that edited your manuscript.A copy of your manuscript showing your changes by either highlighting them or using track changes (uploaded as a *supporting information* file).A clean copy of the edited manuscript (uploaded as the new *manuscript* file).

3. We note that your Data Availability Statement is currently as follows:

“All relevant data are within the manuscript and its Supporting Information files.”

**Additional Editor Comments:**

Minor revision

Based on the evaluation from the reviewer, I recommend minor revisions for this manuscript. Please find the reviewer’s feedback attached.

Reviewers' comments:

Reviewer's Responses to Questions

**Comments to the Author**

1. Is the manuscript technically sound, and do the data support the conclusions?

Reviewer #1: Yes

2. Has the statistical analysis been performed appropriately and rigorously? 

Reviewer #1: Yes

3. Have the authors made all data underlying the findings in their manuscript fully available?

Reviewer #1: Yes

4. Is the manuscript presented in an intelligible fashion and written in standard English?

Reviewer #1: Yes

5. Review Comments to the Author

Reviewer #1: 1. Abstract:

1.1. The result section in abstract: the author should put P-value to confirm the significance at the end of the sentence (Older age, more education, and Job status were associated with higher scores.

2. Introduction: suitable and expressive.

The objectives are clear, and so the presentation

3. Methods: the methods are well described. The author should mention the statistical tests which were used for the data analysis.

4. Results section:

In this part:

4.1. I suggest the authors should define EpiPen for anaphylaxis before the knowledge of using it.

4.2. The authors did not make comparisons of the knowledge score among the participants such as males and females, marital status, education levels, and so on

4.3. The authors did mention the factors affecting the Knowledge, Attitude regarding the awareness of Anaphylaxis.

I suggest the followings:

4.4. The authors should obtain Odd ratios for the factors affecting knowledge of Anaphylaxis among the participants using regression test

4.4.1. The authors should add a regression table to demonstrate the factors that influence the Knowledge or awareness of Anaphylaxis

6. PLOS authors have the option to publish the peer review history of their article (what does this mean?). If published, this will include your full peer review and any attached files.

Reviewer #1: No

---

## [Author Response · Author response to Decision Letter 0]

25 Nov 2024

Dear Editor

We sincerely thank you and the reviewers for their thoughtful and constructive feedback on our manuscript titled "Awareness of Anaphylaxis Among Public in Al-Ahsa City of Saudi Arabia: A Cross-sectional Study." Below, we provide detailed responses to each of the reviewer’s comments and outline the revisions made to the manuscript.

Reviewer Comments and Responses

4.1 Suggestion: The authors should define EpiPen for anaphylaxis before discussing participants' knowledge of its use.

Response: We have modified the manuscript to include a definition of EpiPen in the results section prior to addressing participants’ knowledge of its use, as suggested.

4.2 Suggestion: The authors did not make comparisons of the knowledge score among participants, such as males and females, marital status, education levels, etc.

Response: The difference in mean knowledge scores of anaphylaxis among participants (e.g., males and females, marital status, education levels) has been analyzed and represented in Table 1: Demographic Profiles.

4.3 Suggestion: The authors did mention the factors affecting the knowledge and attitudes regarding awareness of anaphylaxis. The authors should obtain odds ratios for these factors using a regression test.

Response:

• While the primary aim of this study is to assess public awareness of anaphylaxis in the Kingdom of Saudi Arabia (KSA) and calculate the mean awareness score based on survey responses, the secondary objectives do not include identifying factors influencing participants’ knowledge.

• The statistical methods employed were selected based on the nature of the variables. Specifically:

o Chi-square tests were used for bivariate analyses to explore associations between variables.

o For variables with more than two categories, Analysis of Variance (ANOVA) was applied.

o For variables with only two categories, independent t-tests were employed.

• Conducting regression analysis and presenting a regression table to identify factors influencing knowledge is outside the approved objectives and scope of this study.

4.4 Suggestion: The authors should add a regression table to demonstrate the factors that influence knowledge or awareness of anaphylaxis.

Response: The primary aim and statistical framework of this study were focused on descriptive and comparative analyses of awareness scores. Adding regression analyses, including odds ratios and regression tables, would require expanding the study's objectives and revising the methodology. As this falls beyond the scope of the approved research, we respectfully chose not to include regression analyses in the current manuscript.

We hope these responses and revisions address the reviewers' concerns adequately. We are grateful for the opportunity to improve our manuscript and remain available for any further clarifications or additional modifications.

Sincerely,

Dr.Ahmed Alanazy

---

## [Editor Report · Decision Letter 1]

1 Dec 2024

Awareness of Anaphylaxis Among Public in Al-Ahsa City of Saudi Arabia: A Cross-sectional Study

PONE-D-24-36838R1

Dear author,

We’re pleased to inform you that your manuscript has been judged scientifically suitable for publication and will be formally accepted for publication once it meets all outstanding technical requirements.

Kind regards,

Vinh Le Ba, PhD in Pharmaceutical Science

Academic Editor

PLOS ONE
---

## [Editor Report · Acceptance letter]

16 Dec 2024

PONE-D-24-36838R1 

PLOS ONE

Dear Dr. Alanazy, 

I'm pleased to inform you that your manuscript has been deemed suitable for publication in PLOS ONE. Congratulations! Your manuscript is now being handed over to our production team.

Kind regards, 

on behalf of

Dr. Vinh Le Ba 

Academic Editor

PLOS ONE